# A Framework for Comparison and Assessment of Synthetic RNA-Seq Data

**DOI:** 10.3390/genes13122362

**Published:** 2022-12-14

**Authors:** Felitsiya Shakola, Dean Palejev, Ivan Ivanov

**Affiliations:** 1GATE Institute, Sofia University, 125 Tsarigradsko Shosse, Bl. 2, 1113 Sofia, Bulgaria; 2Institute of Mathematics and Informatics, Bulgarian Academy of Sciences, Acad. G. Bonchev St., Bl. 8, 1113 Sofia, Bulgaria; 3Department of Veterinary Physiology and Pharmacology, Texas A&M University, College Station, TX 77843, USA

**Keywords:** simulated data, RNA-seq, differential expression, sample classification, comparative study

## Abstract

The ever-growing number of methods for the generation of synthetic bulk and single cell RNA-seq data have multiple and diverse applications. They are often aimed at benchmarking bioinformatics algorithms for purposes such as sample classification, differential expression analysis, correlation and network studies and the optimization of data integration and normalization techniques. Here, we propose a general framework to compare synthetically generated RNA-seq data and select a data-generating tool that is suitable for a set of specific study goals. As there are multiple methods for synthetic RNA-seq data generation, researchers can use the proposed framework to make an informed choice of an RNA-seq data simulation algorithm and software that are best suited for their specific scientific questions of interest.

## 1. Introduction

RNA sequencing (RNA-seq) technology has revolutionized the way we analyze the dynamic transcriptome. This technology overcomes many of the limitations of microarray platforms—it has the ability to detect unknown genes, is unbiased by antisense probe sequences, has a wider dynamic range to quantify expression and a more advantageous signal-to-noise ratio [1,2]. The most popular applications of bulk RNA-seq include the measurement of gene expression patterns, alternative splicing and isoform expression studies, and the characterization of gene-sets by systems biology approaches. The method has been established as a standard when studying how gene expression is altered by various experimental conditions, diseases or environments. A recent publication provides an overview of new bioinformatics algorithms developed for the purpose of retrieving previously inaccessible information from available RNA-seq data such as: cell type composition (deconvolution), copy number alteration, microbial contamination, and quantification of transposable elements and neoantigen prediction [3].

What must be acknowledged is that bulk RNA-seq does not consider the fact that tissues are composed of various cell types and that there is mounting evidence that gene expression in similar cell types is not homogeneous [4,5,6]. The development of single cell RNA-seq (scRNA-seq) [7,8], especially its high-throughput application [9], has addressed this limitation of bulk RNA-seq and allows for the profiling of cell-to-cell variability on a genome-wide scale. Moreover, the scRNA-seq technology allows for a deeper understanding of diverse biological processes and thus facilitates basic science, as well as clinical research. scRNA-seq has significantly advanced the research on cancer evolution, embriogenesis, stem cell differentiation, and heterogeneity of microbial populations [10,11]. With the advancement of the technology and the signal resolution at the individual cell level, there is an increasing interest in this type of experimental work and in the associated datasets.

Bulk RNA-seq could be used to obtain an estimate of the mean of the gene expression from a cell population for detecting the presence and measuring the quantity of transcripts in a tissue sample at a given moment. This makes bulk RNA-seq best suited to identifying differences between sample conditions. The standard workflow includes wet lab experiments and downstream computational analyses. The wet lab part consists of RNA extraction and isolation, library preparation (RNA fragmentation, nonribosomal RNA enrichment, reverse transcription to cDNA, sequencing adapters ligation, PCR amplification), and sequencing to a read depth of many millions of reads per sample, with the newest technology producing billions of read-pairs per lane on a high-throughput sequencing machine. Bulk RNA-seq usually requires a small initial amount of RNA. The computational analysis part incorporates filtering out low quality reads and adapter sequences, alignment to reference transcriptome, read abundance quantification, gene-based read counting and filtering, and normalization between samples and batches. The relevance of bulk RNA-seq is supported by large public databases (dbGAP, GEO) and its common use in translational research [3].

While scRNA-seq experiments share some laboratory preparation steps with the bulk RNA-seq processing pipeline, they have several specific steps involving single cell dissociation and capture, which is critical for the quality of the obtained data and the downstream analysis. The main technologies of scRNA-seq are plate-, microfluidic- and droplet-based [12]. The computational analysis part in both bulk RNA-seq and scRNA-seq follows similar pipelines. There are different scRNA-seq technologies, with different RNA transcript lengths, numbers of captured cells and read depths per cell [13]. scRNA-seq data are characterized by an excessive amount of zero gene counts (sparsity), which could be either due to the nature of gene expression of single cells (biological) or due to technical reasons (non-biological) [14]. This type of data has further peculiarities including a high level of noise, lower library sizes, large overdispersion, etc., as described in [15].

scRNA-seq data are still not as widely available as bulk RNA-seq data and are much more expensive to produce. The main factors adding to the cost at the time of writing are the costs of the reagents as single cells are processed in individual reactions; the cost of sequencing as many more reads need to be generated; and the technology itself, which requires more research and development [16]. scRNA-seq is rapidly catching up to the bulk RNA-seq in terms of available datasets in various databases (the Single Cell Portal https://singlecell.broadinstitute.org/single_cell (accessed on 8 November 2022), the Hemberg collection [17], SCPortalen [18], and scRNASeqDB [19]), a large number of single cell reference atlases, and cell-querying and annotating algorithms [20]. Recently, attempts have been made to use knowledge from scRNA-seq data for gene signature transfer in bulk RNA-seq transcriptome studies [21].

After the initial computational processing, researchers are most commonly interested in the analysis of differentially expressed genes (DEGs). The main difference in DEGs analysis between bulk RNA-seq and scRNA-seq is that in the case of bulk RNA-seq it is often aimed at differentiating between case vs control conditions (e.g., tumor versus normal), while in the case of scRNA-seq it is aimed at detecting biomarkers across cell types [22,23]. An extended review of the main methodologies for finding DEGs for bulk RNA-seq data is available in [24]. In the case of scRNA-seq, one has to manage a large amount of expression data—the measurements can be for several thousands of genes for many individual cells, generated in a single experiment. Here, the analysis of DEGs is the main tool for the identification of gene markers for cell type detection and for inputs to secondary analyses, such as gene set analysis, gene network and pathways analysis [25]. An operational framework for the analysis of DEGs for scRNA-seq data is presented in [25], along with a classification and summary of the available methods for scRNA-seq data.

The second most common type of analysis aims to discover sets of transcripts that can discriminate between different biological phenotypes, i.e., statistical methods, and supervised or unsupervised machine learning algorithms; thus, these methods attempt to hypothesize about the presence of phenotypes and their relation to experimental conditions or disease states. The common setup of bulk RNA-seq experiments, with samples in the hundreds and measured transcripts in the thousands, leads to the so-called “curse of dimensionality” problem which is further exacerbated by rapidly increasing computational costs. Because of these issues, the analytical approaches are usually coupled with dimensionality reduction techniques [26]. An assessment of how well several supervised methods perform for bulk RNA-seq data is presented in [27]. In the settings of scRNA-seq data, one is usually interested in discriminating either between cell populations or cell types [28]. A number of machine learning-based methods have been developed for these purposes; however, the task of assigning biological functions to clustered or known cell populations remains a challenge [29]. It should be noted that the depth of sequencing is essential for the optimization of the machine learning methods and is strongly dependent on the diversity of cells in the population [30]. With the increase in the amount of available scRNA-seq data, semi-supervised cell type classification methods, utilizing external and well-annotated sources, are becoming popular [31].

Gene expression data can be studied not only at the level of individual genes, but also as gene lists, regulatory pathways or networks because genes interact to support biological processes. Gene regulatory network inference could be based on simple correlations among the measured variables (co-expression) or non-linear associations among subsets of the transcripts represented in the RNA-seq data. Due to the fact that co-expressed genes can often be functionally related, controlled by the same set of transcriptional factors, or part of the same pathway, deciphering co-expression networks can help in clarifying biological modules governing specific biological processes [32]. Interestingly, comparisons between bulk RNA-seq co-expression networks and microarray data-derived networks show much higher correlations in RNA-seq data due to higher sensitivity and a larger dynamic range [33]. Here again, it is also essential to consider how the sample size and the sequencing depth affect the quality of RNA-seq co-expression networks [34]. In order to obtain the “gold standard” co-expression networks, it might be necessary to analyze thousands of samples from different conditions [35].

Additionally, small changes in the expression of some genes may not be detected as statistically significant or informative when each gene is considered in isolation. However, their joint activity could have impactful biological consequences. Therefore, the differential enrichment analysis of biological pathways can sometimes provide a better biological interpretation than focusing on individual genes [36]. An extensive review of available co-expression network- and pathway analyses and other applications of RNA-seq is available here [35]. As the amount of data from scRNA-seq is growing, there is a basis for the development of approaches using partial information decomposition for the detection of regulatory networks [37,38]. There are also attempts to integrate scRNA-seq data with other “omics” datasets for the functional inference of gene regulation [39].

With the increased interest in developing new methods for analysis of bulk and scRNA-seq data, the need for evaluating methodologies and benchmarking the implementations of the associated algorithms also increases. Some of the most popular metrics used for such benchmark performance evaluation are the false discovery rate (FDR) and sensitivity [40], as well as classification error estimation [41]; additionally, clustering [42] and network inference accuracy [43] are important areas of interest. Using real datasets for these types of performance evaluation studies is not reliable because one lacks knowledge of ground truth related to gene expression levels, their differences between populations, their interactions, or even their class label. For example, determining the ground truth may require costly spike-in experiments. Therefore, machine learning and statistical approaches need to be evaluated on datasets where the multidimensional probability distributions are known before deploying them on real datasets. A common cost-effective alternative to using real RNA-seq data for such evaluation purposes is to employ synthetically generated datasets with known generation parameters and a built-in ground truth, resembling real data.

As the amount of transcripts in the RNA-seq raw data is measured in non-negative integer counts, and taking into consideration the systematic biases of the sequencing platforms, the Poisson distribution was first proposed to model RNA-seq data, e.g., in [44]. That distribution has only one parameter and equal mean and variance. This model is not currently recommended due to the overdispersion of real data, usually exhibiting a large read count variation among biological replicates. Therefore, RNA-seq data are now often modeled with two-parameter distributions, such as generalized Poisson distribution or negative binomial distribution [45], that allow more flexibility in modeling. Alternative solutions have been proposed to model the global underlying biological variability, including a tweedy Poisson distribution [46] and a beta-binomial generalized linear model [47], which have certain limitations, as described in [46].

A well-known property of scRNA-seq datasets is that they contain a relatively large proportion of genes with zero counts: the proportion of zeros in bulk RNA-seq data is 10–40% [48,49], while it could reach 90% in scRNA-seq data [50]. This feature of scRNA-seq data creates challenges for the estimation of gene expression correlations [51] and the understanding of gene expression dynamics [52]. The sources of zero counts in scRNA-seq data, along with the impacts of zeros on various data analyses, modeling and interpreting of data and analyses results are discussed in greater detail in [14].

Currently, there are a multitude of scRNA-seq data simulators available, with the majority of them estimating features of a real single-cell dataset in order to generate data. The earliest methods in this direction utilize the negative binomial distribution [49,53,54], or the zero-inflated negative binomial distribution to better model sparsity [55,56]. Other parametric and nonparametric methods are discussed in [57]. Some of the popular current synthetic scRNA-seq data applications could be categorized as follows: (i) studies of the imputation of missing values [58,59], and (ii) cell type identification [60,61,62,63] among others. An overview of the technologies and important problems that could be solved using scRNA-seq is available in [64].

Various synthetic data generators for bulk RNA-seq data have been proposed since the inception of the technology. The early methods simulating bulk RNA-seq data utilize FASTA, SAM or BED files as the input and have goals such as the evaluation of alignment algorithms, studying the experimental biases of the sequencing platforms by mimicking the major experimental steps, or estimation of transcript expression when comparing de novo transcriptome assemblers [65]. Synthetic bulk RNA-seq data generators that have gained more recent popularity are discussed in [66] and they are used predominantly for the purposes of studying differential expression, classification and correlation. Synthetic data are also used in innovative research areas such as for studying k-mer signatures [67], and integration, including with different “omics” technologies, as well as for the normalization of transcriptomic data [68,69,70,71], studying viral diversification dynamics [72], haplotypes [73] and the origin of outbreak [74]. The above list of applications of synthetic RNA-seq data is not exhaustive. As the need for synthetic RNA-seq data and the number of generators are ever-growing, researchers can benefit from a general computational framework that would provide them with an informed choice on whether the intended simulation strategy is optimal.

The following sections describe a general framework, as shown in Figure 1, for the benchmarking of synthetic RNA-seq data generators and provide an example, as shown in Figure 2, of the application of this technique using a few commonly used data generators. It should be noted that this benchmarking example serves as an illustration of the general framework and does not aim at a comprehensive utilization of the synthetic datasets generated for the purposes of this work.

## 2. A General Computational Framework for Selection of Task-Specific Synthetic RNA-Seq Data Generator

As there is a variety of available software aimed at simulating RNA-seq data, it is important to compare performance in a systematic fashion. Our paper introduces a general computational framework that could be used to select a synthetic RNA-seq simulator to generate a sufficient number of samples, appropriate for the evaluation of a specific analysis method or to benchmark several such algorithms against each other, as shown in Figure 1.

When selecting algorithms for benchmarking, one should consider their underlying parametric models for data generation, or lack thereof in the nonparametric case. Parametric approaches use experimental datasets to infer the parameter values for an assumed distribution, whereas nonparametric approaches use experimental datasets as baseline data add add effects to it, and are model-free; therefore, they are considered to be fast [72]. The evolution of the nonparametric approaches has been discussed in [75]. Furthermore, one should consider which of the major goals (DEGs analysis, classification, network analysis, etc.) are addressed by the respective algorithm and its software implementation. Finally, an appropriate analysis of the synthetic data pool should be conducted, for example considering feature-feature correlations, especially when the synthetically-generated data are used to study clustering or the classification of samples or cells.

## 3. Application of the Framework to Benchmark Several Synthetic Bulk RNA-Seq Data Generators

In this article, we focus on one specific application of the proposed computational framework and consider the following important benchmarking criteria in the case of bulk RNA-seq data: (i) distributional characteristics of generated synthetic data; (ii) number of DEGs; and (iii) class-covariance structure, as shown in Figure 2. This example can serve as a guideline for the framework’s application to evaluation and benchmarking tasks for synthetic RNA-seq data generation algorithms and software packages. The software packages that we compare here are aimed at generating synthetic bulk RNA-seq data and take as input either a combination of parameters, a count dataset or both. They have been developed with a focus on the following three major goals: (1) study of DEGs: compcodeR [40], SimSeq [76], powsimR [49], seqgendiff [75]; (2) classification studies: Splatter [54]; and (3) correlation studies: SPsimSeq [77].

Because the software packages of interest were originally designed to have real bulk RNA-seq data as input, we used the AD RNA-seq dataset [78] for comparisons. The cohort is of patients with Alzheimer’s disease and roughly matched controls, without neurological diseases, sequenced on Illumina HiSeq 2500, following a standard polyA-selected Illumina RNA-seq protocol. The data were downloaded from the recount3 repository [79]. After filtering out genes with a median of 0 and subsetting to 50 patients and 50 controls (not necessarily matched, which is the case for most real studies), we obtained comparable datasets for the simulations, with measurements for 34616 genes or transcripts. The data have two classes corresponding to the input requirement for some of the simulators compared in this study, as shown in Figure 2.

We also used synthetically generated RNA-seq data with a known covariance structure (NGSSPPG package, [80,81]) as the input for our benchmarking application. NGSSPPG simulates RNA-seq data with a known covariance structure—predetermined mean, variance and feature correlations. We generated two datasets, NGSSPPG1 and NGSSPPG2, each containing 100 samples with 10,000 genes or features. Both datasets have a clear two-class structure (class ratio = 50/50), and represent the following two distinct levels of difficulty for classification algorithms: a “simple” and easily separable class structure characterized by a small variance parameter, $\sigma_{0}$ = 0.4 (this dataset is denoted by NGSSPPG1 in our study) and a “difficult” mixing class structure characterized by a large variance parameter, $\sigma_{1}$ = 0.7 (this dataset is denoted by NGSSPPG2). The two datasets have a defined feature-feature block correlative structure with a correlation of 0.4 for five features as user-specified variables for the algorithm. As in [66], the mean amount of reads per feature was set to 300 using an empirical procedure accessible in the code (https://github.com/Felitsiya/Comparative-study-of-synthetic-bulk-RNA-seq-generators (accessed on 8 November 2022)), for purposes of comparison with the real RNA-seq data used in our benchmarking, the Alzheimer’s disease (AD) dataset. The NGSSPPG software utilizes the following two-step procedure to generate data: (1) modeling mRNA concentrations with a multivariate Gaussian model (MVN-GC); and (2) modeling NGS-reads using a Poisson process that takes the MVN-GC data as its input. The user can specify the following parameters: number of features, number of samples in each group, number of subgroups in group 1, number of global, heterogeneous and non-markers, average expression and average standard deviation for each group, number of correlated variables in a block structure, strength of block covariance, sequencing depth, and noise.

The procedure we used to simulate data consists of the following steps:(i)We calculated the parameters needed for the benchmarked packages using the NGSSPPG synthetic RNA-seq datasets or the real RNA-seq data (AD dataset) as input for textbfcompcodeR: mean gene expression of class 1, and specific dispersions of class 1 and class 2, as recommended in the package manual. We used the default values for effect size and the minfact and maxfact parameters for the simulated samples’ individual sequencing depths.(ii)**powsimR**’s parameter estimation step was performed with the recommended settings for bulk RNA-seq data; the simulation step was performed with the DESeq2 differential testing method. For this particular simulation, we truncated the values above $10^{7}$ to be equal to $10^{7}$ for the AD dataset, with the goal of avoiding the effect of severe outliers.(iii)As the input of **seqgendiff**, we took either one of the two classes from the NGSSPPG dataset or the control group from the AD dataset. The added signal is from an exponential distribution with a rate of 0.5 and effect size of 1.5.(iv)**SimSeq** runs as a one-step procedure; therefore, we used its default parameters.(v)For **SPsimSeq**, we set the genewiseCor parameter to FALSE; therefore, we chose not to retain the gene-to-gene correlation structure of the input data. This was to avoid the high computational cost of calculating/keeping that structure.

The read count matrices produced by the respective synthetic data generation packages were used for the comparisons, including quantile-quantile (Q-Q) plots [82] and descriptive statistics (countsimQC R package [83]), Figure 2. The numbers of DE genes were determined by DESeq2 [84], Table 1.

Our first comparative metric is based on the Q-Q plots visualization of the similarities between the distributions of the input and the respective outputs of the five benchmarked packages. Q-Q plots illustrate the similarity between two given distributions. If the distributions of two datasets are similar, their quantiles should be close and the scatter plot should be close to the diagonal, as shown in Figure 3. Notably, for the two well-separable classes in the NGSSPPG1 dataset, the Q-Q plots indicate greater similarity between the input data and the output (after using one of the five data simulations methods) data distributions. In contrast, the data with the larger noise component, NGSSPPG2, is not as similar to the datasets simulated by the five packages. The closest distribution pair in these cases is the one with SimSeq, where the regression line is approximately the diagonal. For simulations based on the real RNA-seq data (AD dataset), a larger intrinsic variation is noticeable, which is likely due to the nature of real data and the larger number of features or genes (more than three times as many features than those present in the NGSSPPG datasets), as shown in Figure 3C. The distribution of the real dataset is best matched in the sense of Q-Q plots by the simulated compcodeR dataset.

Our second comparative metric is the Dispersion vs Biological Coefficient of Variation (BCV) plots, as shown in Figure 4. Displayed is the dispersion or so-called biological coefficient of variation versus the mean of the log2 of the counts per million reads, which is calculated using DESeq2 before running the test for detecting DEGs. It is immediately noticeable that the curve that is characteristic for many real datasets, as shown in Figure 4C, is not present in the case of the two-class synthetic RNA-seq data generated by the NGSSPPG package, which can be seen in Figure 4A,B. The observed discrepancy could be a manifestation of the specific goals of the NGSSPPG algorithm, as it aims to generate data for classification purposes, with relatively low dispersion.

The third metric deployed in our benchmarking application of the general framework is based on the mean-variance scatter plots, as shown in Figure 5. These plots show the relationship of the feature variance to their empirical mean, without taking into consideration the experimental design and potential sample grouping. Note that seqgendiff simulates data with dispersion, which are not present in the input data, due to the added predetermined fixed effect size and an extra random variable with an exponential distribution.

The next comparative metric used in our application is based on the Spearman correlation coefficients distribution for pairs of features presented in the feature-feature correlation plots, Figure 6. This figure visualizes the preservation of the input’s correlation structure according to the respective generation algorithms. Only non-constant features are considered—if more than 25 such features are found in a dataset, the pairwise correlations between 25 randomly selected features are displayed [83]. Here, the nonparametric algorithms SimSeq and seqgendiff outperform the other generators in capturing the feature-feature correlations in the real dataset, as shown in Figure 6C. Interestingly, these generators do not perform well in capturing the feature-feature correlations present in the NGSSPPG input.

Another metric used in our benchmarking application is the ability of the synthetic data generators to recapture the number or the percentage of DEGs observed in the input dataset. Note that there is an overwhelming presence of DEGs in the AD dataset, possibly due to the intrinsic nature of the sampled tissues. The synthetic data generators have been set to produce data with 5% DEGs i.e., 500 DEGs for the NGSSPPG-based data and 1731 DEGs for the AD-based data, as shown in Table 1. The numbers of DEGs found in the SimSeq-produced datasets is much lower than expected.

As part of the testing for DEGs, we also generated volcano plots. Volcano plots illustrate the statistical significance of the expression difference relative (y-axis) to the magnitude of difference (x-axis) for every transcript in a comparison between two groups [85]. Figure 7 shows the volcano plots with the greatest resemblance of the input data achieved by SimSeq and SPsimSeq. One can notice that when simulating with real data (AD dataset) as the input, as shown in Figure 7C, that compcodeR, powsimR and seqgendiff produce data, which are skewed towards a positive magnitude of difference, i.e., only overexpression is present, with almost no underexpression between the two groups.

Our final tool for benchmarking the five synthetic bulk RNA-seq data generators is based on principal component analysis (PCA). It demonstrates the separability of groups or classes potentially present in the respective datasets, as shown in Figure 8. Note that PCA cannot completely differentiate the two classes of the AD data, as shown in Figure 8C, possibly due to the large variance present in this particular real dataset. At the same time, the synthetic NGSSPPG generation algorithm provides us with the following two clearly distinct scenarios for class-conditional distributions of the input ground truth data: (i) two clearly separated classes in the NGSSPPG1 dataset, Figure 8A; and (ii) two mixed (because of the large noise or variance) classes in the NGSSPPG2 dataset, as shown in Figure 8B. Note that two of the five benchmarked packages, compcodeR and seqgendiff, produce simulated data that allow PCA to detect two classes as present in the input NGSSPPG datasets. However, seqgendiff produces data with a much stronger separability when compared to both cases of the input NGSSPPG datasets. The possible explanation for this phenomenon could be that seqgendiff adds signal during the process of data generation. This could lead to the potential misrepresentation of the data structure as the comparison of the PCA plots of the AD data vs seqgendiff, which is simulated with it as input, shows.

## 4. Discussion

One of the critical problems related to the evaluation of statistical and machine learning methods for RNA-seq data analyses is the lack of knowledge about the true real data multidimensional distributional properties. Therefore, the generation of synthetic RNA-seq datasets, which can serve as ground truth for such an evaluation, has become a topic of significant interest in the research community. Currently, there is an ever-growing number of methods and software packages for generating synthetic RNA-seq data. Many studies focus on the performance of methods analyzing DEGs, typically comparing the results, e.g., sensitivity, specificity and related performance metrics, and in some cases the statistical power. However, little attention has been paid to using a systematic approach for comparing the quality of the RNA-seq simulated data. In this paper, we propose a general framework to address the problem. We highlight several metrics that can be considered when comparing synthetic RNA-seq data, while also taking into account the properties present in the data used as input for the generation algorithms. The application of the proposed approach is illustrated by using five currently available software packages for the generation of synthetic bulk RNA-seq data. In our opinion, it is important to include both synthetic data (NGSSPPG) with a known structure, and real datasets (AD) in the proposed evaluation. The application of our general computational framework, as shown in Figure 1, to the benchmarking task of comparing these five data generation software packages shows that different synthetic RNA-seq data generators are optimal for capturing different aspects of the input data, such as dispersion, number of DEGs, feature-feature correlations and separability into two classes. SimSeq, seqgendiff and potentially SPsimSeq are best suited for preserving the feature-feature correlation structure while compcodeR appears to be the better choice if the research task is geared towards classification or clustering studies. The observation that different data-generation algorithms perform differently with respect to the proposed application metrics underscores the importance of performing a comparative evaluation before selecting data-generation software in order to evaluate specific data analysis methods. While the steps outlined in Figure 1 and Figure 2 describe the general framework for performance evaluation and its specific implementation in the case of bulk RNA-seq synthetic data generation, it is also evident that more work is needed to expand upon and refine the proposed framework in each particular case, in particular when one needs to find a suitable algorithm and software package for generating synthetic scRNA-seq datasets.

## Figures and Tables

**Figure 1 genes-13-02362-f001:**
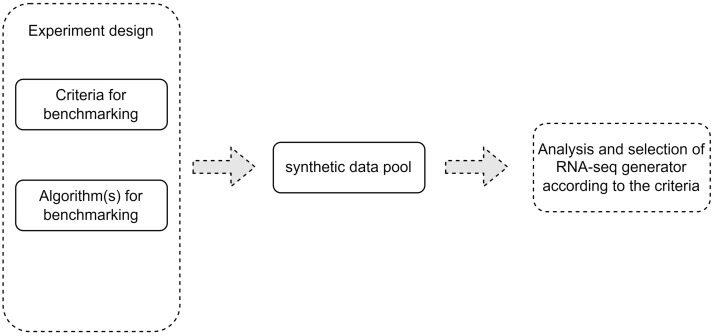
Methodological framework for synthetic RNA-seq data generation for benchmarking of algorithms for statistical and pattern recognition analyses.

**Figure 2 genes-13-02362-f002:**
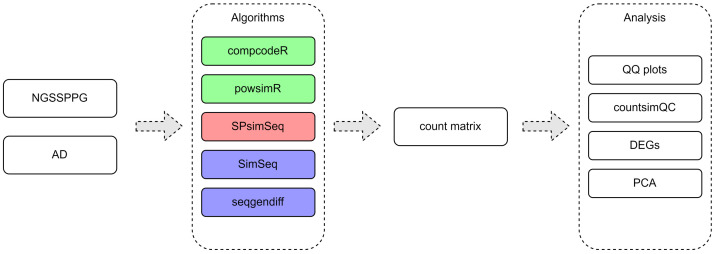
Application of the framework for comparison of bulk RNA-seq generators. Different colors indicate different types of methods. compcodeR and powsimR are parametric, SPsimSeq is semiparametric, and SimSeq and seqgendiff are nonparametric.

**Figure 3 genes-13-02362-f003:**
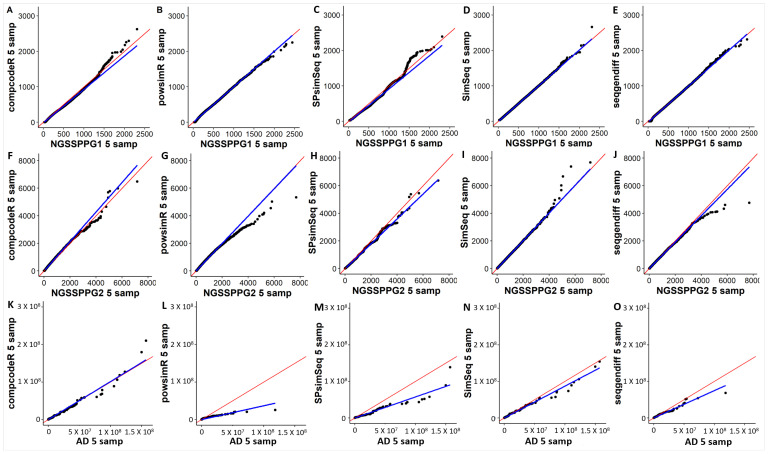
Q-Q plots of five synthetic data samples generated by the respective data generators (y axis) vs. five samples used as input for those data generators (x-axis), with (**A**–**E**) NGSSPPG1 samples used as input data; (**F**–**J**) NGSSPP2 samples as input data; (**K**–**O**) AD samples as input data. The axes represent the quantiles of the respective distributions. Blue: linear regression line. Red: diagonal line.

**Figure 4 genes-13-02362-f004:**
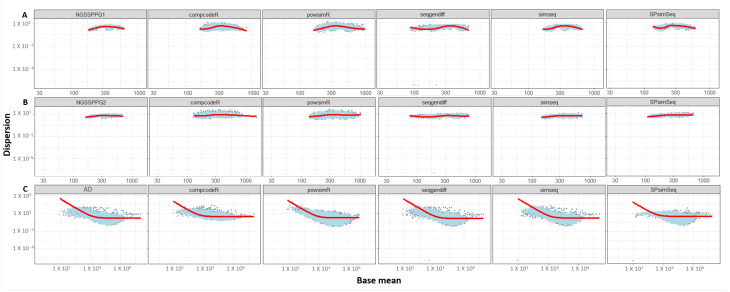
Dispersion vs. BCV plots of the: (**A**) NGSSPPG1; (**B**) NGSSPPG2; and (**C**) AD datasets and the datasets generated with these as input. Black dots: gene-wise dispersion estimates. Red curve: fitted mean-dispersion relationship. Blue circles: final dispersion estimates.

**Figure 5 genes-13-02362-f005:**
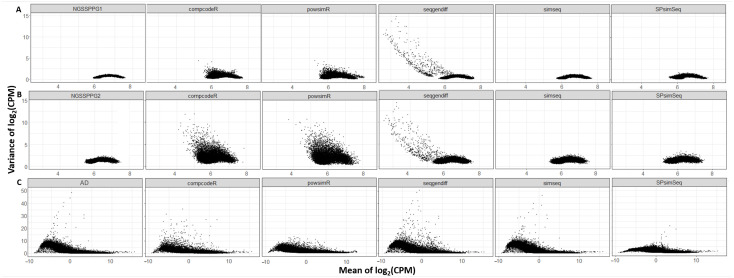
Mean-variance plots of the: (**A**) NGSSPPG1; (**B**) NGSSPPG2; and (**C**) AD datasets and the datasets generated with these as input.

**Figure 6 genes-13-02362-f006:**
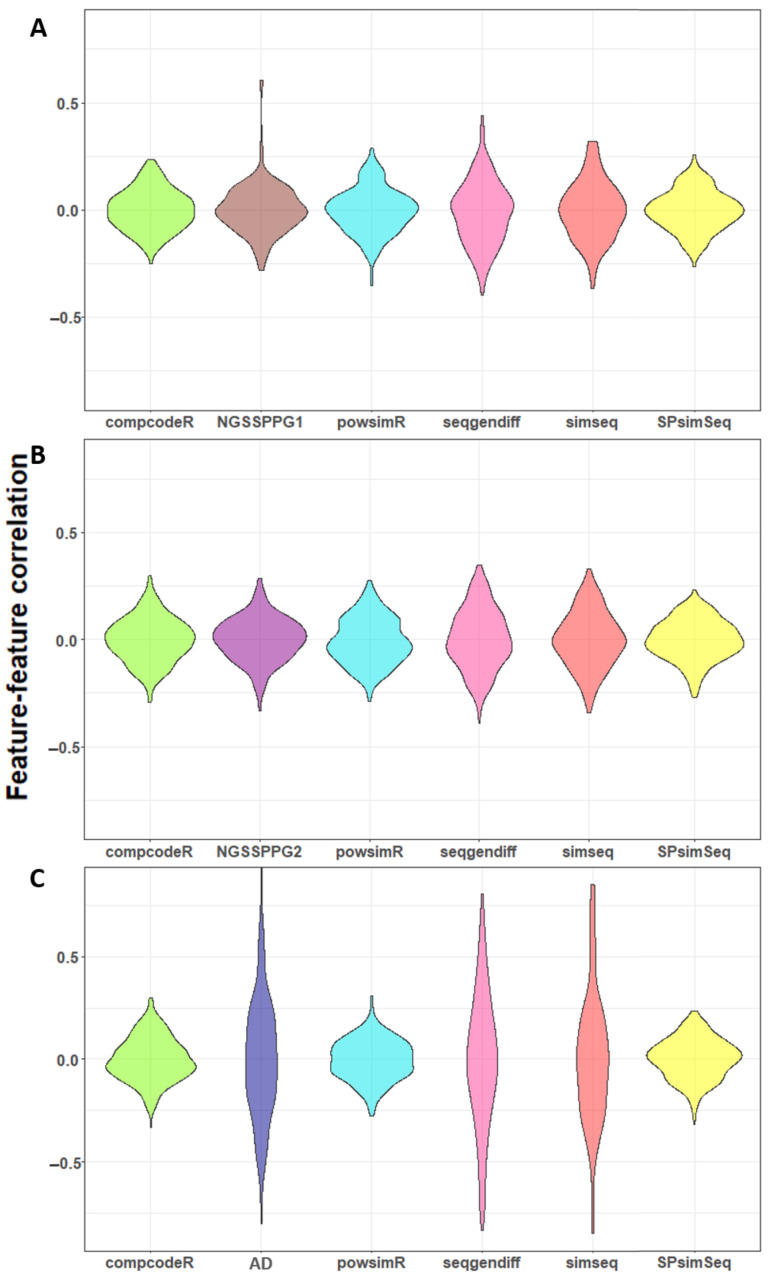
Feature-feature correlation plots of the (**A**) NGSSPPG1; (**B**) NGSSPPG2; and (**C**) AD data and the datasets generated with these as input.

**Figure 7 genes-13-02362-f007:**
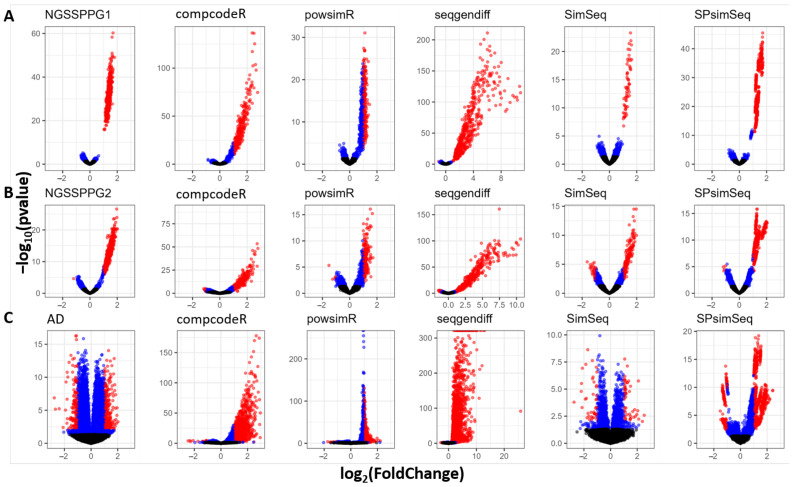
Volcano plots for: (**A**) NGSSPPG1; (**B**) NGSSPPG2; and (**C**) AD datasets and the datasets generated with these as input. Blue dots: transcripts with *p*-value > 0.05, denoting significant differential expression. Red dots: transcripts with *p*-value > 0.05 and $log_{2}\left(FoldChange\right)$ > 1 or with *p*-value > 0.05 and $log_{2}\left(FoldChange\right)$ < −1.

**Figure 8 genes-13-02362-f008:**
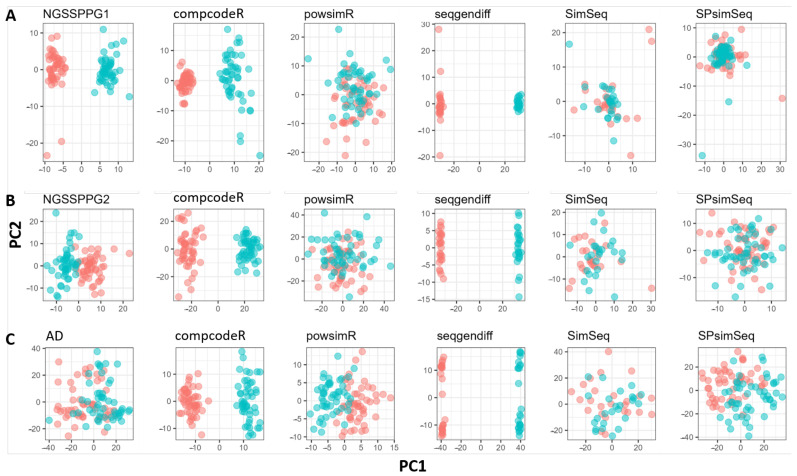
PCA plots for: (**A**) NGSSPPG1; (**B**) NGSSPPG2; and (**C**) AD datasets and the datasets generated with these as input.

**Table 1 genes-13-02362-t001:** DEGs found by DESeq2 from datasets simulated based on either the NGSSPPG1 (100 samples, 10,000 features, class ratio = 50/50) data or the AD data (100 samples, 34,616 features, class ratio = 50/50). All generators are set to produce data with 5% DEGs. Asterisk (*) denotes the original dataset; the subsequent rows indicate datasets generated with it as input.

Dataset	# DEGs	Dataset	# DEGs	Dataset	# DEGs
NGSSPPG1 *	623	NGSSPPG2 *	677	AD *	10066
compcodeR	560	compcodeR	453	compcodeR	1689
powsimR	543	powsimR	449	powsimR	1751
seqgendiff	573	seqgendiff	648	seqgendiff	1649
simseq	77	simseq	238	simseq	297
SPsimSeq	629	SPsimSeq	674	SPsimSeq	1828

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
