# Peer review of "A Framework for Comparison and Assessment of Synthetic RNA-Seq Data"

_genes, 2022, doi:10.3390/genes13122362_

Round 1

Reviewer 1 Report

This is an interesting study and the authors have collected a unique dataset using cutting edge methodology. The paper is generally well written and structured. However, in my opinion the paper has some shortcomings in regards to some data analyses and text, and I feel this unique dataset has not been utilized to its full extent. the introduction needs to give a more specific look of the analysis done later. The github repository provides a well structured information of the dataset used and the analysis. It was not possible to replicate the complete process in given short time but the given code with dataset looked good with initial analysis.

Author Response

Thank you for your thoughtful review.

As suggested, we have added several sentences to the introduction, specifying the analysis done further. Because the main focus of the article is to provide a general framework for studies that can benefit from using synthetic RNA-seq data, we did not include additional comparisons of the various datasets in this work. We are planning to do so in a follow-up article. We have made minor stylistic changes and have fixed some spelling errors. Also we now call the real data set AD, after the Alzheimer’s disease rather than Huntley – the contributing author to the GEO repository.

Reviewer 2 Report

This reviewer commends the authors for a well presented, interesting and potentially useful article.  A short list of editorial replacements follow below:

1) On line 67: "many" instead of "much" 

2) On line 130: "need for" instead of "need of" 

3) On line 182: "number" instead of "amount" 

4) On line 281: "as many" instead of "more" 

5) On line 358: "number" instead of "amount" 

6) On line 360: "best suited" instead of "most suited"

Finally, figure 1 is very sparse. Either add more information to the diagram or eliminate the figure and describe your methodological framework as it is in the text. 

Author Response

Thank you for your thoughtful review.

We followed your suggestions and introduced the respective changes in the article. We opted for keeping Figure 1 as is because it emphasizes the general framework for comparing synthetic RNA-seq data sets for studies/algorithms with different aims. A more detailed content is presented in Figure 2 which illustrates how this general framework is applied to a specific comparative study. We have made some additional minor stylistic changes and have fixed some spelling errors. Also we now call the real data set AD, after the Alzheimer’s disease rather than Huntley – the contributing author to the GEO repository.